# Willingness to join community based health insurance among households in South Wollo, Northeast Ethiopia: A community-based cross-sectional study

**Gebeyaw Biset Wagaw**[1]*, **Abay Woday Tadesse**[2,3,4], **Getahun Yeshiwas Ambaye**[5]

1 Department of Pediatrics and Child Health Nursing, College of Medicine and Health Sciences, Wollo University, Dessie, Ethiopia, 2 Department of Public Health, College of Medicine and Health Sciences, Samara University, Samara, Ethiopia, 3 Dream Science and Technology College, Dessie, Ethiopia, 4 Armauer Hansen Research Institute, Addis Ababa, Ethiopia, 5 Department of Nursing, Bahir Dar Health Science College, Bahir Dar, Ethiopia

* gebeyawbiset@yahoo.com, gebeyaw.biset@wu.edu.et

## Abstract

**Data Availability Statement:** All relevant data are within the manuscript and its Supporting Information files.

### Background

Poor health care financing remains a major challenge to health service utilization among the lower socioeconomic society. Consequently, countries have designed different health insurance programs to overcome financial barriers against health services utilization. Similarly, Ethiopia has been implementing community-based health insurance programs since 2011 to improve health care financing system. However, only a small number of people are enrolled which might be attributed to lack of willingness towards the program and the reasons for this remained under reported. This study was intended to examine willingness to join the community-based health insurance program and its associated factors in South Wollo, Northeast Ethiopia.

### Method

A community-based cross-sectional study was conducted among 421 households. A multi-stage systematic random sampling technique was employed to recruit the study households. Data were entered into EpiData version 3.1 and was exported into SPSS version 24.0 for analysis. Bivariable and multivariable logistic regression analysis with a backward elimination method was performed to identify the determinants of willingness to join community-based health insurance. Finally, a statistically significant level was declared at a p-value of less than 0.05.

### Results

Two hundred and ninety-three [73.6% (95%CI:68.8%-77.9%)] households were willing to join community-based health insurance programs. Being male headed household (AOR:0.2, 95%CI: 0.07–0.58), being a member of Idir (AOR:0.46, 95%CI: 0.25-.84),

**Funding:** The author(s) received no specific funding for this work.

**Competing interests:** The authors have declared that no competing interests exist.

**Abbreviations:** CBHI, Community-Based Health Insurance; CSA, Central Statistical Agency; FMoH, Federal Ministry of Health; UHC, Universal Health Coverage; SHI, Social Health Insurance.

absence of chronic illness in the household (AOR: 0.31, 95%CI: 0.13–0.77), and family size < 4 (AOR: 0.18, 95% CI:0.08–0.41) were barriers to join community-based health insurance program whereas rural residency (AOR:1.9, 95% CI: 1.09–3.32), perceived quality health services (AOR:2.96, 95%CI:1.4–6.24), and having positive attitude (AOR:4.1, 95%CI:2.32–7.22) and good knowledge to programs (AOR:2.62, 95%CI:1.43–4.8) were enabling factors.

## Conclusion

Nearly three-fourths of the households were willing to join community-based health insurance programs. However, different household and health service-related factors affected their willingness. The ministry of health with the regional and woreda health offices should work towards improving the quality of health services, conduct program advocacy and community sensitization towards the program, and build trust with the community.

## Introduction

Globally, the costs of health care are increasing which causes many people to fall into poverty. The out-of-pocket health care expenditures resulted in massive financial barriers to health care utilization among the lower socioeconomic society [1,2]. Every year, nearly 150 million people globally experience financial catastrophe due to out-of pocket health care expenditures. Evidence had suggested that households spend more than 40% of their income on health care services. The catastrophic nature of health care expenditure is disproportionately very high among the rural households in low-income countries where 80% of the health care service is dependent upon out-of-pocket charges. These high health care expenditures cause short-term health shock and can lead to debt and asset sales which intern brings people into deep poverty [3–6].

The world health organization (WHO) recommends different health care financing strategies to reduce the catastrophic nature of out-of-pocket health care expenditures. Subsequently, countries are designing and implementing different health insurance programs. These include community-based health insurance (CBHI), social health insurance (SHI), and private insurance schemes. These health insurance schemes were aimed to provide financial protection against the cost of illness and increase health service utilization. Studies suggested that insured people were more likely to visit health care institutions than uninsured people when they get sick. This suggested that health insurance programs could help in expanding health service utilization and thereby it could help in achieving universal health coverage [7–10].

Universal health coverage (UHC) is a situation in which all individuals and communities receive health services they need without suffering financial hardship [11,12]. High and middle-income countries have achieved UHC through the implementation of health insurance programs [13]. Similarly, developing countries like Ghana, Rwanda, and Nigeria have achieved rapid utilization of UHC through the national health insurance program [14–16]. However, the UHC strategy confronts the challenges of health care financing in developing countries [13–17]. More than half of the African countries are still relying on out-of-pocket health care services. This out of pocket health care expenditure affect health service utilization among the poor rural communities in low income countries which in turn affects the plan of achieving universal health coverage [18–20].

Ethiopia has planned to achieve universal health coverage by 2035 [21]. However, the health service utilization in the country remained low (34.3%) compared to most of the African countries [12]. Studies showed that poor health care financing is a major contributor for the low health service utilization in the country. Consequently, Ethiopia has designed two policy options; community-based health insurance and social health insurance schemes to improve the health care financing system. Community-based health insurance is a voluntary health insurance program designed to improve financial access to health care services for the informal sectors of rural communities. Whereas, social health insurance is a form of mandatory health insurance to the formal sector employees [22,23].

Although the social health insurance proclamation was ratified in 2010, it is not yet implemented in Ethiopia. However, the country has been implementing community-based health insurance programs since 2011. The objectives of launching CBHI were to improve financial access to health care services; increase resource mobilization; and improve the quality of health care services [22,24]. A national pilot study on the effectiveness of CBHI schemes in Ethiopia showed that CBHI members were using health services 26% more than non-members. Additionally, there was a significant difference in the rate of healthcare utilization between insured (50.5%) and uninsured (29.3%) households [10]. This suggested that scaling up community-based health insurance program in the country could help in expanding health service utilization [25].

The health care system in Ethiopia is guided by a 20-year health sector development strategy which is implemented through a series of five-year health sector development programs. Currently, the country is implementing the fourth health sector development plan which has introduced a three-tier health care delivery system. The primary level consists of health posts (1/5,000), health centers (1/25,000), and primary hospitals (1/100,000). Secondary level services are provided by general hospitals (1/1million) and tertiary services by specialized hospitals (1/5million). These three levels of health care systems are integrated through the referral system and provide service for CBHI members. The communities could get their CBHI membership through the health development army and health extension workers at the nearest health center or hospital [26–28].

The community-based health insurance program is organized at the kebele, woreda, regional, and national levels. The woreda administrator is responsible for 'Signing agreements with health care provider's/health facilities; 'Reimbursing health care providers; Administering the fund (keeping financial records; preparing financial statements); Managing the database (which contains data on members, contributions, and utilization). A General Assembly and Board of Directors oversee the governance of CBHI schemes at the woreda level. At the kebele level, the executive body is responsible for registering members, collecting premiums, and channeling funds to each woreda scheme. The communities are directly linked to the program via the leaders of the health development army and the health extension workers [28,29].

The community-based health insurance benefit package includes outpatient and inpatient services, laboratory services, imaging services, supply of drugs, and related services with the exception of eyeglasses, dental implant, dialysis, higher specialized procedures, and aesthetic procedures. All governmental health centers that are situated in the woreda and fulfill the minimum standard of service delivery are contracted to provide services to the members. All pilot woreda has also signed service contracts with their region's respective referral hospitals. Three regions (Amhara, Oromia, and Tigray) entered contracts to ensure the possibility of interregional referrals [28].

Ethiopia has planned to enroll eighty percent of the households under the CHBI program by the end of 2020. However, only 48% of households are utilizing the program with a higher rate of dropout. As a result, the community-based health insurance utilization in the country

had remained one of the lowest in Sub-Sahara African countries. Several factors were responsible for the low achievements of community-based health insurance programs in the country. These factors include educational status, residence, family size of the household, membership in other social supporting systems like Idir, and quality of health service in the catchment area [30,31].

Household willingness is an important determinant for the successful implementation of the CBHI program in the country. However, only a small number (12.8%) of the households were willing to join the program in Ethiopia [32]. Additionally, 36% of the participants who were a member of CBHI are not willing to renew their membership for the next period [29]. This low level of willingness could result in low CBHI utilization and a high dropout rate and which in turn affect the 2035 country plan of achieving universal health coverage [21]. Besides, there was regional variation among studies regarding willingness to join the program and associated factors [33–41]. This study was intended to examine willingness towards the community-based health insurance and associated factors in south wollo zone northeast Ethiopia.

## Methods and materials

### Study area, design, and period

The Community-Based cross-sectional study was conducted in rural households of Wereb-Babu district, South Wollo Zone, Northeast Ethiopia from June to July 2020. South Wollo is one of the 12 zones in the Amhara region with 22 districts which is located 401 km northeast of Addis Ababa, the capital city of Ethiopia, and 480 km to the East of Bahirdar, the capital city of the Amhara region. Based on the population projection of the 2007 national census conducted by the Central Statistical Agency of Ethiopia (CSA), South Wollo has a total population of more than 3 million with an area of 17,067.45 square kilometers. South Wollo has a total of 598,447 households resulting in an average of 4.2 persons to a household [42]. In this study, household heads in 8 kebeles were included, however, household heads residing less than six months in the kebele, critically ill or mentally incapable household heads with no other household members greater than 18 years were excluded (Fig 1).

### Sample size determination

The sample size was determined using a single population proportion formula by considering the 95% confidence interval, margin of error 5% (d = 0.05), proportion (P) = 79% taken from the previous study [31], and 1.5 design effect. The maximum sample size was considered after checking various parameters of measurements from the prevalence and associated factors. Then, the researcher added 10% to compensate for the non-responses and the final sample size for the study was 421.

$$n = \frac{\left(\frac{Za}{2}\right)^2 * p(1-p)}{d^2}$$

$$n = \frac{(1.96)^2 * 0.79(0.21)}{(0.05)^2} \underline{n = 255}$$

$$\underline{n = (255*1.5) + (383*10\%) = 421}$$

Where: n = required sample size, Zα/2 = critical value for normal distribution at 95% confidence level (1.96), p = proportion of CBHI among households, d = 0.05 (5% margin of error), and DEFF = design effect to compensate loss of efficient of sample power.

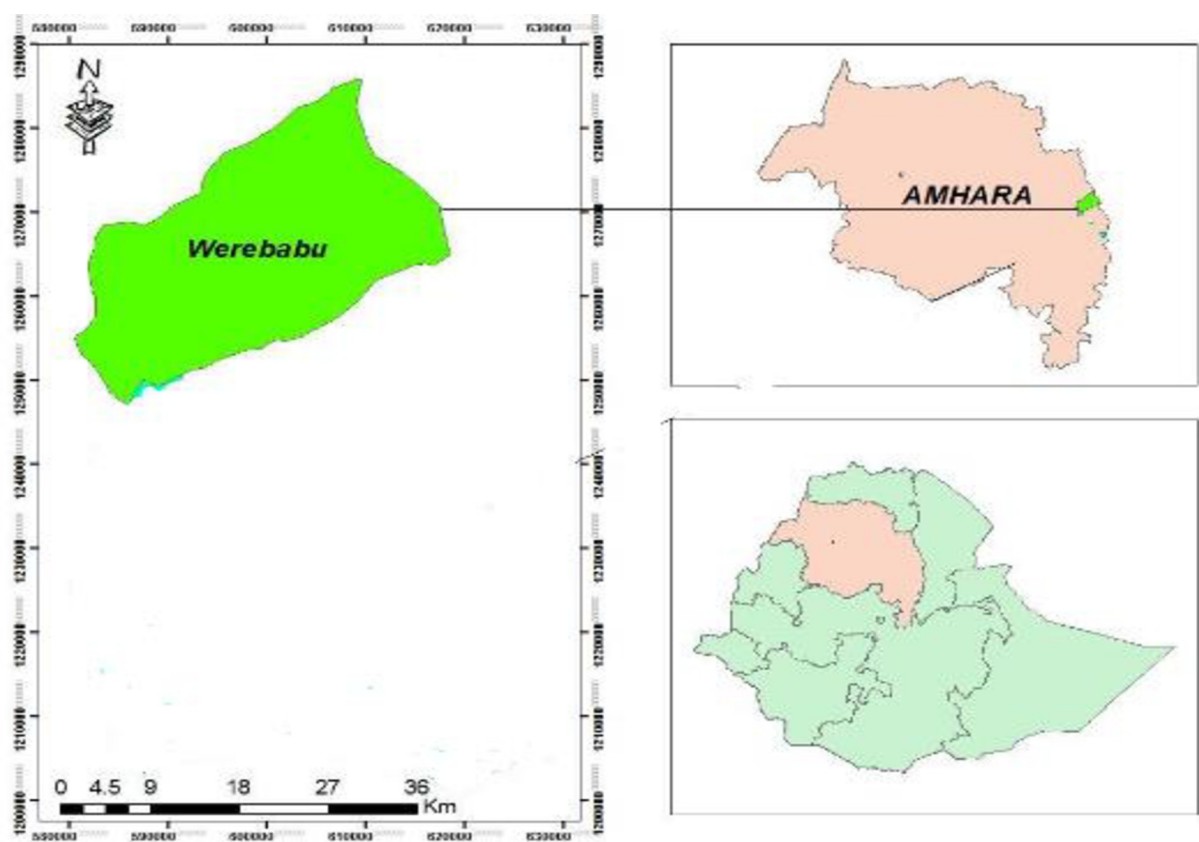

**Fig 1. Map of WerebBabu District, South Wollo Zone, Northeast Ethiopia, July 2020 taken from the Woreda administrative office.**

### Sampling technique

WerebBabu district was selected purposely from the 22 districts found in South Wollo Zone, Amhara regional state. There are 20 kebeles [i.e. lower administrative unit in Ethiopia] in the district. A two-stage sampling technique was used to obtain the study participants. The first 8 kebeles were selected by using the lottery method from 20 kebeles found in the district. Then the study households were selected by systematic random sampling methods from 8 kebeles. The interval was calculated by dividing the total households found in 8 kebeles by the total sample size (i.e. 2842 households divided by 421, which is 7). After obtaining the interval (7), we have selected the index household leveled 5 by lottery method from the 7 households. Then, the study households in 8 kebeles were selected beginning from the indexed household (leveled 5) by systematic random sampling technique in every 7[th] interval till the required sample was reached.

### Data collection tool and procedures

This questionnaire was adapted from different studies conducted in developing countries [40,43,44] and then modified into contexts. The tool consists of; questions to assess sociodemographic status, questions to assess knowledge of the participants, questions to assess attitude of the participant towards the program, and questions to assess willingness towards the community-based health insurance program. The tool was first prepared in English and was translated to Amharic and then back to English to examine its consistency. Finally, the Amharic version questionnaire was used for data collection.

A total of 38 questions were used to assess knowledge towards community-based health insurance programs. Each question was awarded a score of 1 if it was answered correctly and 0 if not answered. Then the total correct responses of each question were summed to yield the knowledge level of the participants. Participants who scored above 50% for the knowledge questions (19 out of 38) were considered to have good knowledge of the scheme whereas respondents who scored below this cut of point were considered to have poor knowledge.

A 10 items of 5 points Likert scale questionnaire were used to assess the attitude of the study participants towards community-based health insurance programs. This was used to represent attitudes to a topic scored on a 5-point scale, i.e. 1 (Strongly Disagree), 2 (Disagree), 3 (Neutral), 4 (Agree) to 5 (Strongly Agree). Participants who scored above the mean were considered to have a positive attitude and those who scored below the mean were considered to have a negative attitude.

Data were collected through face-to-face interviews with the household representatives in each selected household. A total of 8 data collectors with BSc degrees and 4 supervisors with master's degrees were involved in the data collection process. Before the data collection period, 5 days of training were given to the data collectors and supervisors. The data collection process was supervised by principal investigators and supervisors. A week before the actual data collection period a pretest was done on 5% of the total sample size in Dessie Zuriya district.

## Data quality assurance

In this study, a validated tool was used that was adapted from different studies conducted in developing countries including Ethiopia and we modified the tool into context to apply for our study. Pre-testing of the tool was done on 5% of the sample and some amendments were done based on the findings of the pretest. Besides, training was given for data collectors and supervisors regarding the data collection process. Strict supervision on the data collection procedure was held by the principal investigator and supervisors.

The completeness of the data was checked on a daily basis by the supervisors and principal investigators. Data were entered using EpiData version 3.1 data manager and it was checked for completeness, errors, missing values, and then it was exported into SPSS for analysis. The binary logistic regression model was applied to measure the associations of explanatory variables with the outcome variable because the outcome variable in the study is binary and categorical type. Before doing the analysis, the model fitness was checked using the Hosmer-Lemeshow model fit-ness test. Furthermore, the presence of correlation was checked but no collinearity was found.

## Ethics approval and consent to participate

Ethical clearance and approval were obtained from Wollo University. An official letter of cooperation was written to the South Wollo Zone administration and WerebBabu district. Verbal consent was obtained from the study participants after a clear and detailed explanation of the purpose, risks, and benefits of the study.

## Data processing and statistical analysis

The data was entered using EpiData version 3.1 and was exported into SPSS 24.0 for analysis. A bivariate logistic regression analysis was done to assess the association between the dependent variable with each independent variable. The sociodemographic and socio-economic-related factors, knowledge, and attitude-related factors were the independent variables included in the bivariate analysis. Independent variables with a p-value less than 0.2 in the bivariate analysis, clinical importance, and absence of multi-collinearity were considered while

we select the eligible variables for the final model. Multivariable logistic regression analysis was done to control potential confounders and to identify the factors associated with the outcome variable. Finally, a statistical significance level was declared at a p-value of less than 0.05.

### Study variables

**Dependent variable.** Willingness to join community-based health insurance (no = 0, yes = 1)

**Independent variables. Socio-demographic variables:** Age, sex, educational status, family size, Household having children, household having a person above 65 years old age. **Knowledge** (good = greater than mean, poor = less than the mean); **attitude** (positive = greater than mean, negative = less than mean). **Health-related characteristics:** Presence of chronic illness, presence of disability, health institution preference, seeking medical treatment, illness in the past three months. **Health institution characteristics** (health institutions they are served in currently): distance from the household, quality of services, availability of drugs and supplies.

## Results

### Socio-demographic profile of the study participants

A total of 398 households were included with a response rate of 94.5 percent. The mean age of the participants was 41.85(±3.47 SD) years whereas the minimum and the maximum age of the participants were 25 and 63 years respectively. The majority of the participants (83.4%) were male household heads. Regarding their marital and educational status, 87.4% were married and 50% were illiterates (**Table 1).**

### Knowledge toward community-based health insurance

Two hundred and ninety-four (73.1%) participants had good knowledge towards community-based health insurance schemes. Out of these 256 (64.3%) of the participants describe community health insurance programs, 332(83.4%) explain the advantage of community-based health insurance and 324 (81.4%) of the respondents know the health care services under the community-based health insurance program.

### Attitude towards community-based health insurance

The mean attitude of the respondent was 29.7 (±3.7SD) with the minimum and maximum scores of 13 and 37 respectively. The study revealed that 273 (68.6%) of the respondents had a positive attitude towards the community-based health insurance program.

### Willingness to join community-based health insurance

Majority of the households 293 (73.6%) were willing to join community-based health insurance programs. The main reasons for lack of willingness towards the program were lack of interest 19 (18.1%), lack of trust 23 (21.9%), having other means of health care 34 (32.4%), the belief of not getting sick after paying 13 (12.4%), and due to the belief that community-based health insurance causes financial constraints 16 (15.2%).

### Factors associated with willingness to join CBHI program

Male-headed households were 80% [AOR: 0.2, 95%CI:0.07, 0.58] less likely to join community-based health insurance programs. Similarly, participants with a positive attitude towards the CBHI scheme were 4.1 [AOR: 4.1, 95%CI:2.32, 7.22] times more likely to join the

**Table 1. Sociodemographic characteristic of the respondents in WerebBabu district, Northeast Ethiopia, July, 2020.**

| Variable | Category | Frequency(n = 398) | Percentage |
|---|---|---|---|
| **Sex** | Male | 332 | 83.4 |
| | Female | 66 | 16.6 |
| **Age** | 20–29 | 31 | 7.8 |
| | 30–39 | 116 | 29.1 |
| | 40–49 | 169 | 42.5 |
| | >50 | 82 | 20.6 |
| **Marital status** | Single | 11 | 2.8 |
| | Married | 348 | 87.4 |
| | Divorced | 25 | 6.3 |
| | Windowed | 14 | 3.5 |
| **Educational status** | Illiterate | 215 | 54.0 |
| | read and write | 148 | 37.2 |
| | Primary school | 22 | 5.5 |
| | Secondary school | 13 | 3.3 |
| **Residence** | Rural | 314 | 78.9 |
| | Urban | 84 | 21.1 |
| **Family size** | < 3 | 61 | 15.3 |
| | 4–6 | 235 | 59 |
| | > 6 | 102 | 25.6 |
| **Participation in Idir** | Yes | 112 | 28.1 |
| | No | 286 | 71.9 |
| **Seek health care while sick** | Yes | 294 | 73.9 |
| | No | 104 | 26.1 |
| **Chronic illness in the house** | Yes | 32 | 8.0 |
| | No | 366 | 92.0 |
| **Disability in the house** | Yes | 6 | 1.5 |
| | No | 392 | 97.5 |
| **Ever borrowed for health services** | Yes | 133 | 33.1 |
| | No | 265 | 65.9 |

insurance program and respondents with good knowledge of the scheme were 2.6 [AOR: 2.62, 95%CI:1.43, 4.8] times more likely to join the program. Household heads with a family size of less than 4 were 82% [AOR: 0.18, 95%CI:0.08, 0.41] less likely to join the community-based health insurance scheme and respondents who are members of Idir were 54% [AOR: 0.46, 95%CI:0.25, 0.84] less likely to participate in the program. Households who have no chronic illness in their families were 69%(OR:0.31, 95%CI:0.13,0.77) less likely to join the program. Rural households were almost 2 times (OR:1.9, 95%CI:1.09–3.32) more likely to join CBHI program compared to the urban households (**Table 2**).

## Discussion

Community-based health insurance program was designed as a means of financial protection against the cost of health care utilization chiefly among the lower socioeconomic societies. However, studies have suggested that small segments of the population are utilizing the program particularly in developing countries like Ethiopia. Additionally, the community's willingness towards the program is low in Ethiopia compared to most of the African countries and there have been dissimilarity among studies in the country regarding willingness toward the program and associated factors.

**Table 2. Factors associated with willingness to join community-based health insurance program among households in WerebBabu district, Northeast Ethiopia July, 2020.**

| Variables | Category | WTJ | | COR(95% CI) | AOR(95% CI) | p-value |
|---|---|---|---|---|---|---|
| | | No n(%) | Yes n(%) | | | |
| Sex | Male | 98(93.3) | 234(79.9) | 0.28(0.13-.64) | **0.2(0.07–0.58)*** | **0.003** |
| | Female | 7(6.7) | 59(20.1) | 1 | 1 | 1 |
| Residence | Rural | 52(49.5) | 191(65.2) | 1.91(1.22–3) | **1.9(1.09–3.32)*** | **0.02** |
| | Urban | 53(50.5) | 102(34.8) | 1 | 1 | 1 |
| Member of Idir | Yes | 40(38.1) | 72(24.6) | 0.53(0.33–0.85) | **0.46(0.25-.84)*** | **0.01** |
| | No | 65(61.9) | 221(75.4) | 1 | 1 | 1 |
| Quality of health service | Good | 23(21.9) | 84(28.7) | 2.16(1.2–3.9) | **2.96(1.4–6.24)*** | **0.004** |
| | Medium | 37(35.2) | 133(45.4) | 2.13(1.27–3.57) | **2.37(1.23–4.56)*** | **0.01** |
| | Poor | 45(42.9) | 76(25.9) | 1 | 1 | 1 |
| Chronic illness | No | 86(81.9) | 280(95.6) | **0.21(0.1–0.44)** | **0.31(0.13–0.77)*** | **0.01** |
| | Yes | 19(18.1) | 13(4.4) | 1 | 1 | 1 |
| Seek health care service | Yes | 87(82.9) | 207(70.6) | 0.5(0.28–0.88) | 0.59(0.3–1.17) | 0.13 |
| | No | 18(17.1) | 86(29.4) | 1 | 1 | 1 |
| Institution to get Rx | Government | 100(95.2) | 265(90.4) | 0.47(0.19–1.26) | 0.52(0.16–1.75) | 0.29 |
| | Private | 5(4.8) | 28(9.6) | 1 | 1 | 1 |
| Educational level | Illiterate | 60(57.1) | 155(52.9) | 0.47(0.1–2.18) | 0.6(0.13–2.77) | 0.52 |
| | Read/write | 35(33.3) | 113(38.6) | 0.59(0.12–2.78) | 1.33(0.28–6.41) | 0.72 |
| | Primary | 8(7.6) | 14(4.8) | 0.32(0.06–1.8) | 1.21(0.19–7.66) | 0.84 |
| | Secondary | 2(1.9) | 11(3.8) | 1 | 1 | 1 |
| Family size | 1–3 | 37(35.2) | 24(8.2) | 0.27(0.14–0.53) | **0.18(0.08–0.41)*** | **<0.0001** |
| | 4–6 | 38(36.2) | 198(67.6) | 2.2(1.27–3.82) | 1.6(0.85–3.1) | 0.14 |
| | >6 | 30(28.6) | 71(24.2) | 1 | 1 | 1 |
| Attitude | Positive | 44(41.9) | 229(78.2) | 4.96(3.08–7.99) | **4.1(2.32–7.22)*** | **<0.0001** |
| | Negative | 61(58.1) | 64(21.8) | 1 | 1 | 1 |
| Knowledge | Good | 61(20.8) | 232(79.2) | 2.64(1.63–4.27) | **2.62(1.43–4.8)*** | **0.002** |
| | Poor | 43(41.0) | 62(59.0) | 1 | 1 | 1 |

**Key:** * = Significantly associated, 1 = reference, COR = Crude odds ratio, AOR = Adjusted odds ratio, Rx = treatment, WTJ = Willingness to join, CI = confident interval.

This study showed that 73.6% [95%CI:68.8%-77.9%] of the households were willing to join community-based health insurance schemes. The finding was similar to a study in Ethiopia (73%) [45], study in Saudi Arabia (69.6%) [46], and another study in Taiwan (69.5%) [47]. The possible reason might be due to the similarity in the study design or it might be due to similarity in the source populations where both of the studies involve rural households. These low level of community willingness could affect the 2030 plan of achieving universal health coverage.

However, the finding is higher than a study in Malaysia (63.1%) [48], another study in Malaysia (61.1%) [19], and a study in Nigeria (40%) [49]. The possible reason for the discrepancy could be due to the difference in the source populations. The low socioeconomic status of the population in Ethiopia might increase willingness to join community-based health insurance program compared to those higher socioeconomic populations in Malaysia and Nigeria. Besides, differences in sociocultural practice between countries may also explained the discrepancy of the findings.

On the other hand, the finding is lower than studies conducted in Southwest Ethiopia (78%) [44], Northern Ethiopia (79%) [50], northwest Ethiopia, east Gojjam (81.5%) [51],

northwest Ethiopia, Fogera district (80%) [39], Nigeria (98%) [14], western Nigeria (82.45%) [52], and urban Bangladesh (86.7%) [53]. This could be due to the differences in program advocacy or community sensitization towards the community-based health insurance scheme among the study participants or the reason might be due to variation in socioeconomic or sociodemographic characteristics of the study participants.

This study revealed that respondents with good knowledge of CBHI were 2.6 times more likely to join the scheme compared to their counterparts. The finding is similar to a study in Siraro District, Ethiopia [45]. The reason could be knowledge on community-based health insurance gives an understanding about the services under community-based health insurance which enables people to join the insurance program. Similarly, participants having a good attitude on CBHI were 4 times more likely to join the program. This finding is similar to a study in Siraro District, Ethiopia [45]. This could be due to the fact that having a positive attitude toward the community-based health insurance program might increase their willingness to join the program.

Respondents who were a member of Idir were 54% less likely to join a community-based health insurance scheme compared to those who were not a member of Idir. This finding was supported by a study in Siraro District, Ethiopia [45]. The reason could be people having other means of social supporting systems like Idir might not need to join the community-based health insurance scheme due to the belief that they might get help from the social supporting systems during illness.

Male-headed households were 80% less likely to join CBHI compared to female-headed households. The finding is similar to the studies conducted in Ethiopia [51], Kenya [54], and Nepal [55]. This could be male household heads might not face financial constrains compared to the female headed households that could restrain from joining and utilizing the program. Respondents having a small family size (<4) were 82% less likely to join the program. The finding is similar to the studies conducted in Northwest Ethiopia [56] and Southern Ethiopia [32]. This might be justified by people with small family size might have financial potential to cover their health care services using out-of-pocket payments. In addition, households with small family size are less likely to pay health care expenditure compared to households with large family size. Thus, they are less likely to join community-based health insurance program.

In the current finding, rural households were 2 times more likely to join a community-based health insurance program compared to urban households. The finding was inconsistent with the study in Kenya where the urban communities were more likely to join the health insurance program compared to the rural communities [54]. This could be due to the wrong of the urban residencies in Ethiopia about health care services in the community-based health insurance programs. Furthermore, the urban community have been relied on private health facilities where CBHI does not refund their costs after they took the treatments.

Households with no chronic illness among family members were 69% less likely to join CBHI program compared to households with chronic illnesses. The finding is similar to studies conducted in Northwest Ethiopia [56], Southern Ethiopia [32], and Malaysia [48]. The possible reason could be people with no chronic illness in their household might not need frequent health care services and hence paying for the community-based health insurance program is considered as a wastage of resources and might not need to join the program.

Households who perceived good health care services in community-based health insurance were almost 3 times more likely to join community-based health insurance compared to their counterparts. The finding was similar to the studies conducted in Northeast Ethiopia [56] and Saudi Arabia [46]. This could be due to the fact that good quality health services satisfy the community and increase their willingness to participate in the community-based health insurance program.

### Implication of the study to the households, health system, and policy

In this study, three-fourths (73.6%) of the households are willing to join CBHI program. This level of willingness might affect the 2020 country plan of achieving 80% of enrollment and the 2035 plan of achieving universal health coverage in the country. The study revealed that the community might have concerns about the program like the quality of health services and lack of trust. Therefore, the government should build trust among the community, increase the quality of health services, and address other household and healthcare-related factors to scale up the program.

### Limitation of the study

Although the authors had tried to maintain the quality of the study, our finding is not without limitations. The major limitation of the study was the cross-sectional nature of the study which is weak to explore determinants of willingness towards community-based health insurance programs. The other limitation of the study might be social desirability bias as data collectors were health care professionals they might affect the study.

## Conclusion

Nearly three fourth of the participants were willing to join a community-based health insurance program. Several household and health service-related factors affected the community's willingness toward the program. The government the regional and local health offices should improve the quality of health services, perform program advocacy (improve knowledge and attitude), and build trust among the community to scale up the program.

### Areas to be explored in the future

The major research areas we recommend for future researchers are; first community willingness with a pure qualitative studies are recommended to explore the deep-rooted barriers for the community to join community based health insurance program. Second, the quality of health care services under community-based health insurance programs should be assessed as well as regular monitoring and evaluation should be done. Thirdly, community awareness creation and community sensitization should be done to improve community willingness toward the program. Additionally, a comparative study on the health service utilization among community-based health insurance members and nonmembers should be conducted.

## Supporting information

**S1 Dataset.**
(SAV)

## Acknowledgments

### Declarations

We are highly indebted to Dream Science and Technology College and Wollo University for permitting us to conduct the study and providing the necessary preliminary information while conducting this study. We would also like to extend our appreciation to the study participants, supervisors, and data collectors.

## Author Contributions

**Conceptualization:** Gebeyaw Biset Wagaw, Abay Woday Tadesse, Getahun Yeshiwas Ambaye.

**Data curation:** Gebeyaw Biset Wagaw, Getahun Yeshiwas Ambaye.

**Formal analysis:** Gebeyaw Biset Wagaw, Abay Woday Tadesse, Getahun Yeshiwas Ambaye.

**Investigation:** Gebeyaw Biset Wagaw, Abay Woday Tadesse, Getahun Yeshiwas Ambaye.

**Methodology:** Gebeyaw Biset Wagaw, Abay Woday Tadesse, Getahun Yeshiwas Ambaye.

**Project administration:** Abay Woday Tadesse, Getahun Yeshiwas Ambaye.

**Software:** Gebeyaw Biset Wagaw, Abay Woday Tadesse, Getahun Yeshiwas Ambaye.

**Supervision:** Gebeyaw Biset Wagaw, Getahun Yeshiwas Ambaye.

**Validation:** Gebeyaw Biset Wagaw, Abay Woday Tadesse, Getahun Yeshiwas Ambaye.

**Visualization:** Gebeyaw Biset Wagaw, Getahun Yeshiwas Ambaye.

**Writing – original draft:** Gebeyaw Biset Wagaw, Abay Woday Tadesse, Getahun Yeshiwas Ambaye.

**Writing – review & editing:** Gebeyaw Biset Wagaw, Getahun Yeshiwas Ambaye.

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
