## [Decision Letter · Decision Letter 0]

26 Jan 2021

PONE-D-20-29657

Willingness to Join Community Based Health Insurance among Households in Worebabo District, South Wollo Zone, Northeast Ethiopia

PLOS ONE

Dear Gebeyaw Bisat

Thank you for submitting your manuscript to PLOS ONE. After careful consideration, we feel that it has merit but does not fully meet PLOS ONE’s publication criteria as it currently stands. Therefore, we invite you to submit a revised version of the manuscript that addresses the points raised during the review process.

General comments: The paper addresses important areas of health system to contribute to sustainable health service delivery.

Introduction : The introduction lacks detail review of literature and policy context of the study country.

Methods: the sample size calculation assumes high rate of enrollment while the result was low which indicates poor review of evidences including the use of estimate from policy document, not peer reviewed paper. This impacts on the power of the study which will not measure the evidence if at all it exists. in addition, the tools used for data collection, how it was developed, pretesting of tools, quality assurance, when data was collected and by who was not clear. the study lacks operational definition for some items like knowledge, attitude etc.

Discussion: the discussion is more of results and the comparison with other studies was random does not consider the similarity among settings and justify why the difference was observed.

Literature review: needs review for more literature and country policy context and provisions. Please find reviewer 's comment included.

We look forward to receiving your revised manuscript.

Kind regards,

Daniel Gemechu Datiko

Academic Editor

PLOS ONE

Additional Editor Comments:

General comments: The paper addresses important areas of health system to contribute to sustainable health service delivery.

Introduction : The introduction lacks detail review of literature and policy context of the study country.

Methods: the sample size calculation assumes high rate of enrollment while the result was low which indicates poor review of evidences including the use of estimate from policy document, not peer reviewed paper. This impacts on the power of the study which will not measure the evidence if at all it exists. in addition, the tools used for data collection, how it was developed, pretesting of tools, quality assurance, when data was collected and by who was not clear. the study lacks operational definition for some items like knowledge, attitude etc.

Discussion: the discussion is more of results and the comparison with other studies was random does not consider the similarity among settings and justify why the difference was observed.

Literature review: needs review for more literature and country policy context and provisions.

Journal Requirements:

2. Please provide further information regarding the pre testing and validation of the survey or questionnaire used in the study and ensure that you have provided sufficient details that others could replicate the analyses. If the questionnaire is not under a copyright more restrictive than CC-BY, please include a copy, in both the original language and English, as Supporting Information.

Furthermore, please provide additional information about the participant recruitment method, for instance inclusion and exclusion criteria applied during the study. Please ensure you have provided sufficient details to replicate the analyses.

3. Please include your tables as part of your main manuscript and remove the individual files. Please note that supplementary tables should be uploaded as separate "supporting information" files.

5. Thank you for submitting the above manuscript to PLOS ONE. During our internal evaluation of the manuscript, we found significant text overlap between your submission and the following previously published works, some of which you are an author.

-Community-based health insurance and social capital: a review (https://doi.org/10.1186/2191-1991-2-5).

-WILLINGNESS-TO-PAY FOR COMMUNITY BASED HEALTH INSURANCE BY FARMING HOUSEHOLDS: A CASE STUDY OF HYGEIA COMMUNITY HEALTH PLAN IN KWARA STATE, NIGERIA (DOI: 10.15547/tjs.2016.03.014).

-DETERMINANTS OF WILLINGNESS TO JOIN COMMUNITY- BASED HEALTH INSURANCE SCHEME IN A RURAL COMMUNITY OF NORTH-WESTERN NIGERIA (10.35202/AJHE.2015.5103).

Please revise the manuscript to rephrase the duplicated text, cite your sources, and provide details as to how the current manuscript advances on previous work. Please note that further consideration is dependent on the submission of a manuscript that addresses these concerns about the overlap in text with published work.

Reviewers' comments:

Reviewer's Responses to Questions

**Comments to the Author**

1. Is the manuscript technically sound, and do the data support the conclusions?

Reviewer #1: Partly

2. Has the statistical analysis been performed appropriately and rigorously? 

Reviewer #1: No

3. Have the authors made all data underlying the findings in their manuscript fully available?

Reviewer #1: No

4. Is the manuscript presented in an intelligible fashion and written in standard English?

Reviewer #1: No

5. Review Comments to the Author

Reviewer #1: Comments

This study assessed willingness to participate in the community-based health insurance and its associated factors in one of the districts in the Northeastern part of Ethiopia. The objective was clearly stated and the authors used quantitative data so as to investigate the issue in question. This shows the study strong.

Despite its good parts the manuscript has many shortcomings.

#1. Similar studies were conducted in different parts of Ethiopia so far. Please follow the following link to access some of them. What the authors brought was not stated in the article.

https://www.ncbi.nlm.nih.gov/pmc/articles/PMC6337798/

https://www.ncbi.nlm.nih.gov/pmc/articles/PMC4074337/

https://www.ncbi.nlm.nih.gov/pmc/articles/PMC6980847/

https://www.ncbi.nlm.nih.gov/pmc/articles/PMC7493329/

#2. Was CBHI scheme already started in Worebabo district? This issue should be clearly stated in the manuscript. The readers have to read the whole result in reference to the availability/non-availability of the scheme in the district.

#3. The paper has a significant write-up and grammatical problem. Therefore, it needs copy edit by a native English speaker.

Detailed points

Abstract

#1. Better if the authors try to minimize judgmental statements if it is not part of their result. For instance in the first page of the manuscript the authors stated that “Community-based health insurance scheme is the best means of strategy…”

#2. The reason and the conclusion stated in the introduction of the abstract doesn’t seem have a logical relationship. What formation is lacking should be the reason for this kind of studies.

Introduction

#1. The introduction is not strong enough to state the problem and convince the readers. The authors have not given strong justification for the study. Therefore, the authors need to enrich the introduction and show what is really lacking?

You may use the following references:

https://www.ncbi.nlm.nih.gov/pmc/articles/PMC5797722/

https://www.ncbi.nlm.nih.gov/pmc/articles/PMC6089309/pdf/bmjopen-2017-019613.pdf

https://pubmed.ncbi.nlm.nih.gov/15315121/

https://www.ncbi.nlm.nih.gov/pmc/articles/PMC4602341/

https://www.ncbi.nlm.nih.gov/pmc/articles/PMC4191130/

#2. Some of the information look contradictory with the existing reality in Ethiopia and have not been cited. For instance: “Ethiopia is one of the high burden countries for both communicable and non-communicable diseases in Africa.” “…only practiced by some of the urban communities…”

#3. Evidence indicate that the CBHI scheme in Ethiopia focuses on the rural community rater that the urban (https://link.springer.com/article/10.1007/s10754-005-2333-y). However, in this article the authors stated that the services are being better used by the urban community. It doesn’t seem reasonable argument.

Methods

Study design

#1. The authors need to define what kebele is in its first appearance.

Sample size determination and sampling procedure

#2. The authors taken 80% of power, even though we don’t need powers to calculate samples in a single population proportion formula. Better if the authors indicate the actual formula (at least citation) they have used during sample size determination.

#3. Using a proportion of 79% of enrolment doesn’t seem logical despite we have lots of evidence on willingness to join CBHI. https://www.ncbi.nlm.nih.gov/pmc/articles/PMC6337798/;
https://www.ncbi.nlm.nih.gov/pmc/articles/PMC4074337/;
https://www.ncbi.nlm.nih.gov/pmc/articles/PMC6980847/;
https://www.ncbi.nlm.nih.gov/pmc/articles/PMC7493329/.

#4. How 8 kebeles were selected from the 20?

#5. Further explanation on the sampling technique is needed. It cannot be clearly understood how the researcher actually gone through systematic random sampling, for instance; from where they got the list of households, total number of households, the sampling interval, from which household they have started the data collection.

#6. The last sentence “Then the study households were selected by systematic random sampling method from 8 Kebeles after proportional allocation.” S not clear for understanding. Would you please revise it?

Data collection tool and procedures

#7. Please indicate and cite from where the authors adapted the tool.

#8. As indicated by the authors the tool consists of items to assess: sociodemographic status of the participants, knowledge of the participants towards CBHI, and community’s attitude towards the program. However, the main outcome variable (willingness to join) was not listed here. I recommend the authors to submit both versions (English and the translated) of the data collection tool as a supplementary file.

#9. The authors wrote that “It was first prepared in the English language and then translated to Amharic to see for consistency.” But is not clear that how one check consistency of a tool translating into another language.

#10. The authors didn’t say anything about the validation and reliability of data collection tools. These information are paramount important in such kind of scientific investigations.

#11. Even the authors didn’t say anything about pretest of the data collection tools.

#12. The authors need to show what were the variables and their measurement.

Data processing and statistical analysis

#14. The authors shown that, graphs were one method to present the findings of the study. However, there are no graphs throughout the document.

#15. Do the authors have any reason to conduct a simple logistic regression to all the variables and select those with p – value < 0.2 for the multiple logistic regression analysis?

#16. A fitness of good test result is not available in the study. Would you please show how the final model fits?

Results

#1. The calculated sample size were 421, similarly the respondents were 421. Therefore, how the response rate be 94.5 percent?

#2. How can we understand illiterate? Is it to mean those who have net attended formal education?

#3. The statement “Participants knowledge was assessed using 38 knowledge questions with two alternatives coded as 1 for correct answers and 0 for incorrect answers. Respondents who scored above 50% (19 out of 38) were categorized as having good knowledge.” Better be part of the methods.

#4. The statement “Out of these 256 (64.3%) of the participants describe community health insurance program, 332(83.4%) explain the advantage of community-based health insurance and 324 (81.4%) of the respondents know the health care services under the community-based health insurance program.” Are not clear. How the authors brought these information?

#5. This statement “Participant’s attitude towards the CBHI scheme was assessed using the 10 Likert Scale questionnaires. The total response of the participant for these 10 questions was summed which provide a minimum of 10 and a maximum of 50 scores. Respondents who scored above the mean score were categorized having a positive attitude towards CBHI program.” Should be part of methods.

#6. The statement “In this study, willingness to join a Community-Based Health Insurance scheme was determined if all the families were willing to join.” Gives an impression that you interviewed all the family members in the selected households. Was the study conducted that way?

#7. The authors should re-write to clarify and define all the variables listed as reasons for not being willing “The main reasons for unwillingness to join community-based health insurances membership were lack of interest in the program (6.3%), the belief of being a member when needed (7.3%), having other means of health care (10.1%), don’t believe in paying for the sickness (4.5%) and due to the belief community-based health insurance causes financial constraints (5%).”

#8. The authors should also make clear that how the reasons of not being willing to join calculated, what was the denominator. Better if they provide the details in a table.

#9. These statement “Bivariable and multivariable analysis of binary logistic regression model was carried out to identify the factors associated with willingness to join community-based health insurance scheme. Variables with a p-value of less than 0.2 in bivariable analysis was entered to multivariable analysis and those variables with a p-value of less than 0.05 in multivariable analysis was considered statistically significant.” In page 6 is a part of methods section and is repetition in the results section.

#10. Have the authors checked the assumptions of binary logistic regression? For instance, there is a cell value less 5, (table 2 variable: educational level).

Discussion

The discussion should be framed in such a way that you analyze every key finding in the light of findings of earlier researchers. Simply providing others findings and what we found is not helpful to analytically present the findings. Just give a highlight of the findings and give discussion of that finding – its interpretation, implication, comparison with earlier findings. After comparison indicate possible reasons for disparity and/or similarity.

6. PLOS authors have the option to publish the peer review history of their article (what does this mean?). If published, this will include your full peer review and any attached files.

Reviewer #1: No

---

## [Author Response · Author response to Decision Letter 0]

20 Apr 2021

Point-by-point Responses to the editors 

Introduction: The introduction lacks detail review of literature and policy context of the study country.

Response: The authors exhaustively work on the introduction section and we had incorporated all of the concerns raised by the editor in the revised version of the manuscript. Besides, we included the literatures to show the policy context of the country, Ethiopia. 

Methods: the sample size calculation assumes high rate of enrollment while the result was low which indicates poor review of evidences including the use of estimate from policy document, not peer reviewed paper. This impacts on the power of the study which will not measure the evidence if at all it exists. in addition, the tools used for data collection, how it was developed, pretesting of tools, quality assurance, when data was collected and by who was not clear. the study lacks operational definition for some items like knowledge, attitude etc.

Responses: The authors have tried to find all the available literatures on willingness to join community-based health insurance program. However, the proportion 79% were used for sample size determination. Because the prevalence reported in this study participants have similar socio economic characteristics with our study communities.

Regarding the tool, data collection procedure, quality assurance, pretesting, and data collector; the authors have revised the comment and incorporated the raised issues in the revised version of the manuscript. 

Discussion: the discussion is more of results and the comparison with other studies was random does not consider the similarity among settings and justify why the difference was observed.

Response: The authors have revised the discussion section and incorporated the concerns of the editor in the revised version of the manuscript.

Literature review: needs review for more literature and country policy context and provisions. Please find reviewer 's comment included.

Response: The authors have tried to review more literatures and country policy context and provision 

Responses to the Reviewer #1

This study assessed willingness to participate in the community-based health insurance and its associated factors in one of the districts in the Northeastern part of Ethiopia. The objective was clearly stated and the authors used quantitative data so as to investigate the issue in question. This shows the study strong.

Despite its good parts the manuscript has many shortcomings.

#1. Similar studies were conducted in different parts of Ethiopia so far. Please follow the following link to access some of them. What the authors brought was not stated in the article.

https://www.ncbi.nlm.nih.gov/pmc/articles/PMC6337798/

https://www.ncbi.nlm.nih.gov/pmc/articles/PMC4074337/

https://www.ncbi.nlm.nih.gov/pmc/articles/PMC6980847/

https://www.ncbi.nlm.nih.gov/pmc/articles/PMC7493329/

Response: Sure, there are some studies conducted in Ethiopia but the previous studies were conducted in places where the households live in non-drought affected areas. However, there was no study conducted in South Wollo Zone that were frequently affected by drought. Besides, these part of the zone, Worebabo, is also affected by desert locust. These factors may affect the willingness of the households to join CBHI. Therefore, these were the reasons to conduct this study in south wollo zone. 

#2. Was CBHI scheme already started in Worebabo district? This issue should be clearly stated in the manuscript. The readers have to read the whole result in reference to the availability/non-availability of the scheme in the district.

Response: The program was already started in worebabo district but still many households were not joined the CBHI scheme and are not utilizing it.

#3. The paper has a significant write-up and grammatical problem. Therefore, it needs copy edit by a native English speaker.

Response: The authors have corrected the typographical errors in the revised manuscript.

Detailed points:

Abstract

#1. Better if the authors try to minimize judgmental statements if it is not part of their result. For instance, in the first page of the manuscript the authors stated that “Community-based health insurance scheme is the best means of strategy…”

Responses: The authors have corrected such judgmental statements in the revised manuscript.

#2. The reason and the conclusion stated in the introduction of the abstract doesn’t seem have a logical relationship. What formation is lacking should be the reason for this kind of studies.

Introduction

Response: We have corrected it, see the revised manuscript.

#1. The introduction is not strong enough to state the problem and convince the readers. The authors have not given strong justification for the study. Therefore, the authors need to enrich the introduction and show what is really lacking?

Response: The previous studies were conducted in places where the households live in non-drought affected areas. However, there was no study conducted in South Wollo Zone that were frequently affected by drought. Besides, these part of the zone, Worebabo, is also affected by desert locust. These factors may affect the willingness of the households to join CBHI. Therefore, these were the reasons to conduct this study in south wollo zone. 

#2. Some of the information look contradictory with the existing reality in Ethiopia and have not been cited. For instance: “Ethiopia is one of the high burden countries for both communicable and non-communicable diseases in Africa.” “…only practiced by some of the urban communities…”

Response: This concern is well addressed in the revised version of the main manuscript.

#3. Evidence indicate that the CBHI scheme in Ethiopia focuses on the rural community rather that the urban (https://link.springer.com/article/10.1007/s10754-005-2333-y) . However, in this article the authors stated that the services are being better used by the urban community. It doesn’t seem reasonable argument.

Response: In our study urban community stands for people who are living in the woreda, kebele, and other small cities in the study area which are also the targets for community based health insurance program. Rural community in this study mean people living in a scattered manner other than the cities mentioned. 

Methods

 Study design

#1. The authors need to define what kebele is in its first appearance.

Sample size determination and sampling procedure

Response: There are 20 kebeles in the study district, Worebabo woreda, of these kebeles, 8 were selected by lottery method. Then study households were selected using systematic random sampling method after proportional allocation for the 8 selected kebeles. The interval (7) was obtained by dividing the total source population to the sample size (2842/421 = 7). Then the index household was selected by lottery method from 1 to 7 values which was 5, the starting from the fifth household we move on the 7th interval until we reached the required sample size. Here is a schematic presentation of sampling techniques.

THY = total households in yaya, KNSHY = number of sample households in yaya

THG = total households in goha, KNSHG = number of sample households in goha

THB = total households in bokekisa, KNSHB = number of sample households in bokekisa

THb = total households in Bulbulo, KNSHb = number of sample households in Bulbulo

THD = total household’s in Deye, KNSHD = number of sample households in Deye

THE = total households in Ejiressa, KNSHE = number of sample households in Ejiressa

THG = total households in Gerebabo, KNSHG = number of sample house holdes in Gerebabo

THg = total households in gedero, KNSHg = number of sample house holdes in gedero

#2. The authors taken 80% of power, even though we don’t need powers to calculate samples in a single population proportion formula. Better if the authors indicate the actual formula (at least citation) they have used during sample size determination.

Response: The sample size was calculated in two ways one using prevalence and second using the associated factors. Then, the maximum sample size was considered for this study.

Objective 1: prevalence of outcome variable and the sample size was determined using;

(1) a single population proportion formula by considering the assumption Zα/2=critical value for normal distribution at 95% confidence level which equals to 1.96 (z value at α=0.05), P (Estimated proportion) =79% taken from previous study in Kewiot and Efratana Gedem Districts of Amhara Region, Ethiopia and absolute precision or margin of error 5% (d = 0.05). The following formula was used to calculate sample size [43].

n=((Za/2)^2*p(1-p))/d^2 

n=((1.96)^2*0.79(0.21))/〖(0.05)〗^2 n = 255

n = 255*1.5) + (382*0.1) = 421

Objective 2: factors associated with the outcome variable and the sample size was determined using double population proportion formula and this was done using Epinfo version7.0.2 with the following basic assumptions

(2) using the assumption Zα/2=critical value for normal distribution at 95%, 80% power of the study, and unexposed to exposed ration of 1 using Epi Info version 7 software. Two variables, knowledge about CBHI and family health status were taken from the previous studies for the sample calculation [39]. Hence power of the study was used while calculating sample size by factors. 

Finally, the maximum sample size was used for this study. By default, the prevalence gives the maximum sample size than the factors. But the 80% power was not determined for the prevalence study.

#3. Using a proportion of 79% of enrolment doesn’t seem logical despite we have lots of evidence on willingness to join CBHI. https://www.ncbi.nlm.nih.gov/pmc/articles/PMC6337798/;
https://www.ncbi.nlm.nih.gov/pmc/articles/PMC4074337/;
https://www.ncbi.nlm.nih.gov/pmc/articles/PMC6980847/;
https://www.ncbi.nlm.nih.gov/pmc/articles/PMC7493329/.

Response: From the retrieved articles the prevalence 79% was near to the study area and can better represent our population. So we assume that the socio economic status in both setting is similar and hence we used this proportion for sample size calculation.

#4. How 8 kebeles were selected from the 20?

Response: 8 kebeles were selected from the 20 kebeles by lottery method (see the main manuscript).

#5. Further explanation on the sampling technique is needed. It cannot be clearly understood how the researcher actually gone through systematic random sampling, for instance; from where they got the list of households, total number of households, the sampling interval, from which household they have started the data collection.

Response: From a total of 20 kebeles in the district 8 kebeles was selected using lottery methods. The total number of households was retrieved from each the selected kebele administration. Then the total sample size was distributed proportionally to each study kebeles based on the number of households in each kebeles. The total source populations were divided by the total sample size to obtain the interval (k=7). After getting the interval (7) one house hold was selected by lottery method from 1-7 kebeles which was the fifth household. Then starting from the fifth household move on every 7th interval to get the total sample size in each kebeles.

Key

Selected kebeles b = bulbulo kebele, D=deye, G =gedero Y=yaya, g=goha, G=gerbabo,

B = bokekisa, E =ejierssa.

THY–total households in yaya KNSHY-number of sample households in yaya; k=613/76=7

THG-total households in goha KNSHG-number of sample households in goha; k=265/39=7

THB-- total households in bokekisa KNSHB-number of sample households in bokekisa; 220/33=7

THb- total households in Bulbulo, KNSHb-number of sample households in Bulbulo; k=418/62=7

THD- total household’s in Deye, KNSHD-number of sample households in Deye, k=338/50=7

THE- total households in Ejiressa, KNSHE-number of sample households in Ejiressa; k=256/38=7

THG- total households in Gerebabo, KNSHG-number of sample house holdes in Gerebabo; k=466/69=7

THg- total households in gedero, KNSHg-number of sample house holdes in gedero; 367/54=7

Total households of 8 kebeles=2842. Total sample of households of 8 kebele= 421; k=2842/421=7

#6. The last sentence “Then the study households were selected by systematic random sampling method from 8 Kebeles after proportional allocation.” S not clear for understanding. Would you please revise it?

Response: Revised 

Data collection tool and procedures

#7. Please indicate and cite from where the authors adapted the tool.

Response: cited (see the revised manuscript).

#8. As indicated by the authors the tool consists of items to assess: sociodemographic status of the participants, knowledge of the participants towards CBHI, and community’s attitude towards the program. However, the main outcome variable (willingness to join) was not listed here. I recommend the authors to submit both versions (English and the translated) of the data collection tool as a supplementary file.

Response: The participants were asked whether they are willing to participate or not in CBHI program after detailed explanation of the aim, payment of the program.

#9. The authors wrote that “It was first prepared in the English language and then translated to Amharic to see for consistency.” But is not clear that how one check consistency of a tool translating into another language.

Response: Translation was made by different scholars and see inconsistencies between translations and made correction to the disagreement.

#10. The authors didn’t say anything about the validation and reliability of data collection tools. This information is paramount important in such kind of scientific investigations.

Response: the tool was adapted from the literatures and no need of validating it but reliability was done via pretest. 

#11. Even the authors didn’t say anything about pretest of the data collection tools.

Response: Pretest was done on 5% of the total sample size (22 study participants) in Dessie Zuriya district. Based on the result of pretest, some ambiguous questionnaire was modified to obtain better clarity to participants.

#12. The authors need to show what were the variables and their measurement.

Response: Corrected (see the revised manuscript)

Data processing and statistical analysis

#14. The authors shown that, graphs were one method to present the findings of the study. However, there are no graphs throughout the document.

Response: corrected (see the revised manuscript).

#15. Do the authors have any reason to conduct a simple logistic regression to all the variables and select those with p – value < 0.2 for the multiple logistic regression analysis?

Response: We did not have special reasons but we consider p-value<0.2 to make the model more relaxed and to include more variable for the final model.

#16. A fitness of good test result is not available in the study. Would you please show how the final model fits?

Response: Hosmer and Lemeshow fitness of good test was done which was 0.87 this implies the model is fitted.

Results

#1. The calculated sample size was 421, similarly the respondents were 421. Therefore, how the response rate be 94.5 percent?

Response: The sample size was 421 with 398 participants responded giving a response rate of 94.5%

#2. How can we understand illiterate? Is it to mean those who have net attended formal education?

Response: In this study, illiteracy mean participants who have not attended formal education.

#3. The statement “Participants knowledge was assessed using 38 knowledge questions with two alternatives coded as 1 for correct answers and 0 for incorrect answers. Respondents who scored above 50% (19 out of 38) were categorized as having good knowledge.” Better be part of the methods.

Response: We have moved this part to the method section

#4. The statement “Out of these 256 (64.3%) of the participants describe community health insurance program, 332(83.4%) explain the advantage of community-based health insurance and 324 (81.4%) of the respondents know the health care services under the community-based health insurance program.” Are not clear. How the authors brought these information?

Response: Participants were asked to answer: what is CBHI?, what is the advantage of CBHI?, what are the health care services covered by CBHI? And the response were recorded 

#5. This statement “Participant’s attitude towards the CBHI scheme was assessed using the 10 Likert Scale questionnaires. The total response of the participant for these 10 questions was summed which provide a minimum of 10 and a maximum of 50 scores. Respondents who scored above the mean score were categorized having a positive attitude towards CBHI program.” Should be part of methods.

Response: We have moved this part to the method section.

#6. The statement “In this study, willingness to join a Community-Based Health Insurance scheme was determined if all the families were willing to join.” Gives an impression that you interviewed all the family members in the selected households. Was the study conducted that way?

Response: No, we have interviewed the head of the house hold members about the willingness of self, willingness of all family members and willingness of some family members.

#7. The authors should re-write to clarify and define all the variables listed as reasons for not being willing “The main reasons for unwillingness to join community-based health insurances membership were lack of interest in the program (6.3%), the belief of being a member when needed (7.3%), having other means of health care (10.1%), don’t believe in paying for the sickness (4.5%) and due to the belief community-based health insurance causes financial constraints (5%).”

Response: Corrected (See the revised manuscript) 

#8. The authors should also make clear that how the reasons of not being willing to join calculated, what was the denominator. Better if they provide the details in a table.

Response: In our study 105(26.4%) of the households were not willing to join the program. The reason for not willingness were calculated among those who were not willing to join the community based health insurance program which was the denominator. The denominator was 105.#9. These statement “Bivariable and multivariable analysis of binary logistic regression model was carried out to identify the factors associated with willingness to join community-based health insurance scheme. Variables with a p-value of less than 0.2 in bivariable analysis was entered to multivariable analysis and those variables with a p-value of less than 0.05 in multivariable analysis was considered statistically significant.” In page 6 is a part of methods section and is repetition in the results section.

Response: We have moved this part to the method section.

#10. Have the authors checked the assumptions of binary logistic regression? For instance, there is a cell value less 5, (table 2 variable: educational level).

Response: The authors have checked the assumption of binary logistic regression

Discussion

The discussion should be framed in such a way that you analyze every key finding in the light of findings of earlier researchers. Simply providing others findings and what we found is not helpful to analytically present the findings. Just give a highlight of the findings and give discussion of that finding – its interpretation, implication, comparison with earlier findings. After comparison indicate possible reasons for disparity and/or similarity.

Response: the authors had revised the whole section of the discussion and incorporated the raised concerns in the revised version of the manuscript. 

Thank you all for your constructive comments as well as for suggesting important literatures!

---

## [Editor Report · Decision Letter 1]

5 May 2021

PONE-D-20-29657R1

Willingness to Join Community Based Health Insurance among Households in Worebabo District, Northeast Ethiopia: A community based study.

PLOS ONE

Dear Gebeyaw Biset 

Thank you for submitting your manuscript to PLOS ONE. After careful consideration, we feel that it has merit but does not fully meet PLOS ONE’s publication criteria as it currently stands. Therefore, we invite you to submit a revised version of the manuscript that addresses the points raised during the review process.

1 - the background needs description about UHC and its relation with CBHI in Ethiopia

2 - implication of the study to households, health system and policy makers is not clear

3 - what does high level of willingness mean to the CBHI in Ethiopia and compare with the current practice and use

4 - it is not clear what strategies should be used for different population groups if we have to enhance participation in CBHI

5 - limitation of the study did not address key issues related to data collection and quality assurance. For example some socially desirable questions and their answers might have affected the results

We look forward to receiving your revised manuscript.

Kind regards,

Daniel Gemechu Datiko

Academic Editor

PLOS ONE
---

## [Author Response · Author response to Decision Letter 1]

15 May 2021

Point by point responses 

1 - the background needs description about UHC and its relation with CBHI in Ethiopia

Responses: Universal health coverage (UHC) is a situation in which all individuals and communities receive the health services they need without suffering financial constraints when paying for their health care services. This can be achieved when everyone has access to health care at an affordable cost [11, 12]. However, UHC strategy confronts the challenges of health care financing, resource allocation and protecting people against financial hardship particularly in developing countries [13]. Following this, countries have been implementing different health care financing systems to reduce the cost of health care expenditures and achieving universal health coverage [14-17]. 

Although Ethiopia has planned to achieve universal health coverage by 2035 [], its UHC has remained low (34.3%) compared to most of the African countries [12]. As a result, the government of Ethiopia has been implementing CBHI program as a means of achieving universal health coverage in the country [21]. A national pilot study on the effectiveness of CBHI schemes in Ethiopia showed that CBHI members are using health services more (26% or more) than the non-members. This implies that universal health coverage can be achieved in the country through the implementation of CBHI program [22]. � The raised concerns are well addressed in the revised version of the manuscript.

2 - implication of the study to households, health system and policy makers is not clear

Response: 

Implication of the study to the households, health system, and policy

In this study, three-fourths (73.6%) of the households are willing to join CBHI program. This level of willingness might affect the 2020 country plan of achieving 80% of enrollment and the 2035 plan of achieving universal health coverage in the country. The study revealed that the community have concerns toward the program like quality of health services and lack of trust to the program. Therefore, the government should build trust among the community, increase the quality of health care system, and address other household and health system related factors to scale up the program.

3 - what does high level of willingness mean to the CBHI in Ethiopia and compare with the current practice and use

Response: Ethiopia has planned to enroll 80% of its population in to the community based health insurance program by 2020. To achieve this plan, more than 80% of the population should be willing to join the CBHI program. Compared to this country plan, studies reported willingness more than 80% were considered high level of willingness and a willingness report below this percent is said to be low level of willingness. But, as we noted from the literature, although there is a regional variation regarding the willingness towards CBHI program, there is a study which revealed only 12.8% of the participants were willing to join the program which was far from the 2020 country plan.

4 - it is not clear what strategies should be used for different population groups if we have to enhance participation in CBHI

Response: In our study we have assessed community’s willingness towards the community based health insurance program in general using similar tool. We have not seen willingness across the different segment of the population. But the finding implies considering the family size, the quality of health services in the catchment area, trust to the program, presence of other social support system in the area should be considered.

5 - limitation of the study did not address key issues related to data collection and quality assurance. For example, some socially desirable questions and their answers might have affected the results.

Response: Corrected (See the revised manuscript)

---

## [Editor Report · Decision Letter 2]

24 May 2021

PONE-D-20-29657R2

Willingness to Join Community Based Health Insurance among Households in South Wollo, Northeast Ethiopia: A community-based study.

PLOS ONE

Dear Gebeyawu Biset,

Thank you for submitting your manuscript to PLOS ONE. After careful consideration, we feel that it has merit but does not fully meet PLOS ONE’s publication criteria as it currently stands. Therefore, we invite you to submit a revised version of the manuscript that addresses the points raised during the review process.

**Abstract **

In the conclusion, instead of directly linking to health extension program it is better to highlight how this can be done from the current CBHI. Health extension program mainly provides preventive and limited curative service which may require insurance coverage.

**Background **

Please add few paragraphs about social health insurance before diving into CBHI

Please comment about the willingness to pay as the CHI system is government led which may affect the willingness. Please add some comments to limitation of the study.

**Methods  **

Please add a paragraph or more about data quality assurance from its preparation to analysis.

**Discussion   **

What is those with fewer family size less likely to participate? Is there anything that the CBHI amount is set based on family size?

Is willingness affected by social desirability of the response? Do we lack some information as the country plans future scale up?

**Conclusion: **Revise the conclusion only based on the study results not the conviction of researchers.

We look forward to receiving your revised manuscript.

Kind regards,

Daniel Gemechu Datiko

Academic Editor

PLOS ONE

Journal Requirements:

Additional Editor Comments (if provided):

Abstract

In the conclusion, instead of directly linking to health extension program it is better to highlight how this can be done from the current CBHI. Health extension program mainly provides preventive and limited curative service which may require insurance coverage.

Background

Please add few paragraphs about social health insurance before diving into CBHI

Please comment about the willingness to pay as the CHI system is government led which may affect the willingness. Please add some comments to limitation of the study.

Methods

Please add a paragraph or more about data quality assurance from its preparation to analysis.

Discussion

What is those with fewer family size less likely to participate? Is there anything that the CBHI amount is set based on family size?

Is willingness affected by social desirability of the response? Do we lack some information as the country plans future scale up?

Conclusion

Revise the conclusion only based on the study results not the conviction of researchers.

---

## [Author Response · Author response to Decision Letter 2]

4 Jun 2021

Point by point response 

Abstract 

#1 In the conclusion, instead of directly linking to health extension program it is better to highlight how this can be done from the current CBHI. Health extension program mainly provides preventive and limited curative service which may require insurance coverage.

Response: In this study, nearly three-fourths of the participants were willing to join a community-based health insurance program. However, different household and health service-related factors were affecting their willingness. Improving the quality of health services, building good knowledge and positive attitude among the community increase the likely of enrollment towards the program. Therefore, the government in collaboration with the regional and woreda health offices should work towards improving the quality of health services and conduct program advocacy via integrating the program with the health extension program. The raised comments are incorporated in the main document.

Background 

#2 Please add few paragraphs about social health insurance before diving into CBHI

Response: Poor health care financing remains a major challenge for the health care system in Ethiopia leaving households vulnerable to impoverishment from catastrophic health expenditures. Community based health insurance and social health insurance are the two policy options designed to address the poor health care financing in the country. Community-based health insurance is an alternative to user fees to improve equity in access to medical care particularly to those rural communities and the informal sectors. Whereas social health insurance is a form of mandatory health insurance for formal sector employees, including retirees and pensioners. Community-based health insurance program has been implemented in the country since 2011. Although Ethiopia take initiatives to implement the social health insurance program, the program is not yet started in the country. The raised comments are incorporated in the main document.

#3 Please comment about the willingness to pay as the CHI system is government led which may affect the willingness. Please add some comments to limitation of the study. Subsidize.

Response: The majority of the health service provisions in Ethiopia are government run. This is also good opportunity to apply CBHI scheme as the insurance schemes are not primarily for profit and require high government subsidy at their very start. However, if the community-based health insurance were privatized, it could be primarily for profit organizations which might expose the community for further costs. 

Methods 

#4 Please add a paragraph or more about data quality assurance from its preparation to analysis.

Response:

Data quality assurance:

A validated tool was adapted from the literature and contextualized into the study area. Prior to the data collection period a pretest was done and based on the result of the pretest some amendments were done. Training was given for data collectors and supervisors regarding the data collection process. The completeness of the data was checked on daily basis by the data collector itself, supervisors and principal investigators. Finally, data was checked for completeness, errors, missing values, and then it was coded entered using EpiData version 3.1 data manager. The binary logistic regression model was selected for analysis because the outcome variable in the study is binary and categorical type. Before doing the analysis, the model fitness was checked using the Hosmer-Lemeshow model fit-ness test which was 0.87. additionally, the correlation between independent variables was checked and there was no correlation between independent variables. The raised comments are incorporated in the main document.

Discussion 

#5 What is those with fewer family size less likely to participate? Is there anything that the CBHI amount is set based on family size?

Response: CBHI payment is based on the family size of the household. In our study families with family size of fewer than 4 were less likely to join the program and the finding was supported by other two studies in the country. And we have justified the reason for this as; households with small family size could afford health cost for their small family and might not financial hardship for health care services as a result they might not need to join the program compared to households with larger families. 

#6 Is willingness affected by social desirability of the response? Do we lack some information as the country plans future scale up?

Response: Willingness might be affected by the social desirability of the response; people might be voluntary to join the program when an interviewer whom they know asked them about their willingness. As the interviewer were health care professionals, the interviewee might be falsely willing to join because they feel fear the health care professional if they say I am not willing to join the program. But, later on they might not pay the premium because they don’t want to join the program. 

#7 Conclusion: Revise the conclusion only based on the study results not the conviction of researchers.

Response: In this study, nearly three fourth of the participants were willing to join a community-based health insurance program. Program. Several household and health service related factors were affecting community’s willingness toward the program. The government should improve the quality of health services and perform program advocacy to scale-up the community based health insurance program (see the revised manuscript).

---

## [Editor Report · Decision Letter 3]

18 Jun 2021

PONE-D-20-29657R3

Willingness to Join Community Based Health Insurance among Households in South Wollo, Northeast Ethiopia: A community-based study.

PLOS ONE

Dear Gebeyawu,

Thank you for submitting your manuscript to PLOS ONE. After careful consideration, we feel that it has merit but does not fully meet PLOS ONE’s publication criteria as it currently stands. Therefore, we invite you to submit a revised version of the manuscript that addresses the points raised during the review process.

Thanks for addressing major comments raised are well addressed. However, the paper will benefit from some additions indicated below

Map of the study areaDescription of the health system including the community structureThe organization of CBHI and what service it covers, and how it works with the community in terms of participation and useAdd areas to be explored in the future research

We look forward to receiving your revised manuscript.

Kind regards,

Daniel Gemechu Datiko

Academic Editor

PLOS ONE
---

## [Author Response · Author response to Decision Letter 3]

19 Jul 2021

Point by point response

1. Map of the study area

Response: (see the main document)

2. Description of the health system including the community structure

Response:

Ethiopian health system is guided by a 20-year health sector development strategy implemented through a series of five-year health sector development programs (HSDP) with the alignment of international commitments. Currently, the country is implementing the fourth health sector development plan (HSDP IV) which has introduced a three-tier health care delivery system in the country. The primary level consists of health posts (1/5,000 population), health centers (1/25,000), and primary hospitals (1/100,000); secondary level services are provided by general hospitals (1/1million population); and tertiary services by specialized hospitals (1/5million populations). The three health care systems are integrated by the referral system where the community are referred from lower level of health care to a higher level and vis versa. The lower levels of health care (health post, health center and general hospitals) are the first entry point for the community. The community are organized in to health development army and they are integrated with the health extension workers so as to enable the community to take greater responsibility for promoting and maintaining their own health. 

3. The organization of CBHI and what service it covers, and how it works with the community in terms of participation and use

Response: 

The community based health insurance is organized in to kebele level, district level, regional and national levels by administration. At the woreda level, this body is responsible for: `Signing agreements with health care provider’s/health facilities; `Reimbursing health care providers; Administering the fund (keeping financial records; preparing financial statements); Managing the database (which contains data on members, contributions, and utilization). A General Assembly and Board of Directors oversee the governance of CBHI schemes at the woreda level. At the kebele level, the executive body is responsible for registering members, collecting premiums, and channeling funds to each woreda scheme. the community are directly linked to the service via the health development army and health extension workers. 

The CBHI benefit package includes outpatient and inpatient services, laboratory services, imaging services, supply of drugs and related services with the exception of eyeglasses, dental implant, dialysis, etc. All government health centers that are situated in the woreda and fulfill the minimum standard of service delivery are contracted to provide services to members. All pilot woreda have also signed service contracts with their region’s referral hospitals. Three regions (Amhara, Oromia, and Tigray) entered contracts to ensure the possibility of interregional referrals.

4. Add areas to be explored in the future research

Response: 

The major research areas we recommend for future researchers are; first community willingness with a qualitative research design need to be under taken in order to get a deeper insights regarding their willingness. Second the researchers need to focus on the health care system including the quality of health care particularly quality of health services under community based health insurances. Additionally, a comparative study regarding health seeking behaviors between community based health insurance utilizers and non-utilizers need to be conducted.

---

## [Editor Report · Decision Letter 4]

21 Sep 2021

PONE-D-20-29657R4Willingness to Join Community Based Health Insurance among Households in South Wollo, Northeast Ethiopia: A community-based study.PLOS ONE

Dear Gebeyaw Biset,

Thank you for submitting your manuscript to PLOS ONE. After careful consideration, we feel that it has merit but does not fully meet PLOS ONE’s publication criteria as it currently stands. Therefore, we invite you to submit a revised version of the manuscript that addresses the points raised during the review process.

The paper has significantly improved through the revision process. However, the discussion  section is more of results and needs detailed discussion of the results. In addition, the areas of focus for future research are too general and do not clearly indicate what should be done. 

Please submit your revised manuscript by November 12, 2021. If you will need more time than this to complete your revisions, please reply to this message or contact the journal office at plosone@plos.org. Please include the following items when submitting your revised manuscript:A rebuttal letter that responds to each point raised by the academic editor and reviewer(s). You should upload this letter as a separate file labeled 'Response to Reviewers'.A marked-up copy of your manuscript that highlights changes made to the original version. You should upload this as a separate file labeled 'Revised Manuscript with Track Changes'.An unmarked version of your revised paper without tracked changes. You should upload this as a separate file labeled 'Manuscript'.If applicable, we recommend that you deposit your laboratory protocols in protocols.io to enhance the reproducibility of your results. Protocols.io assigns your protocol its own identifier (DOI) so that it can be cited independently in the future. For instructions see: https://journals.plos.org/plosone/s/submission-guidelines#loc-laboratory-protocols. Additionally, PLOS ONE offers an option for publishing peer-reviewed Lab Protocol articles, which describe protocols hosted on protocols.io. Read more information on sharing protocols at https://plos.org/protocols?utm_medium=editorial-email&utm_source=authorletters&utm_campaign=protocols.

We look forward to receiving your revised manuscript.

Kind regards,

Daniel Gemechu Datiko

Academic Editor

PLOS ONE

Journal Requirements:

Additional Editor Comments (if provided):

The paper has significantly improved. However, the discussion is more of results and needs more description, comparing the results with other studies and its practical implication. In addition, the areas for future research are too general and does not indicate what should be done.

Reviewers' comments:

The paper has significantly improved through the revision process. However, the discussion  section is more of results and needs detailed discussion of the results. In addition, the areas of focus for future research are too general and do not clearly indicate what should be done. 

---

## [Author Response · Author response to Decision Letter 4]

20 Nov 2021

Response to the editor 

# 1: The discussion section is more of results and needs detailed discussion of the results

Response: we have corrected the discussion section of the manuscript (see the main manuscript).

#2: The areas of focus for future research are too general and do not clearly indicate what should be done

Response: the authors have tried to clarify the area of focus for future research (see the main manuscript)

The authors have also assessed all the necessary editorial problems and corrected it accordingly.

---

## [Editor Report · Decision Letter 5]

9 Dec 2021

Willingness to Join Community Based Health Insurance among Households in South Wollo, Northeast Ethiopia: A community-based cross-sectional study.

PONE-D-20-29657R5

Dear Gebeyaw Bisnat,

We’re pleased to inform you that your manuscript has been judged scientifically suitable for publication and will be formally accepted for publication once it meets all outstanding technical requirements.

Kind regards,

Daniel Gemechu Datiko

Academic Editor

PLOS ONE

---

## [Editor Report · Acceptance letter]

19 Jan 2022

PONE-D-20-29657R5 

Willingness to Join Community Based Health Insurance among Households in South Wollo, Northeast Ethiopia: A Community-based Cross-sectional Study 

Dear Dr. Biset:

I'm pleased to inform you that your manuscript has been deemed suitable for publication in PLOS ONE. Congratulations! Your manuscript is now with our production department. 

Kind regards, 

on behalf of

Dr. Daniel Gemechu Datiko 

Academic Editor

PLOS ONE